# Assessing Impact of Land Use Change on the Ecosystem Service Value in Yinchuan City from 1980 to 2018

**Bo Wang [1,2,*] and Taibao Yang [1]**

1   College of Earth and Environmental Sciences, Lanzhou University, Lanzhou 730000, China; yangt@lzu.edu.cn
2   Department of Design and Art, North Minzu University, Yinchuan 750000, China
*   Correspondence: wangbo8774@163.com

**Abstract:** Accurate assessment and response analysis of land use and land cover change (LUCC), and ecosystem service values (ESV), are critical to regional ecological security and economic development. There is a lack of detailed reports on the impact of LUCC on the temporal and spatial evolution of ESV in Yinchuan City, which is inconsistent with the pilot urban design policy. This paper, using the LUCC data of Yinchuan City from 1980 to 2018, calculated the ESV, analyzed the temporal and spatial patterns of LUCC and ESV, and discussed the response of ESV to LUCC. The results show that, from 1980 to 2018, the building land increased significantly in Yinchuan City, as did the cultivated land. Meanwhile, grassland and bare land decreased, while forest and the water body remained stable. The spatial connectivity of the building land showed regular improvements, while the urban landscape developed in a regular and balanced direction. During the study period, the total ESV of Yinchuan City decreased by $0.75 \times 10^9$ yuan. This was due to the decrease in grasslands and the increase in building area. The supply, regulation and support of three types of services have a high correlation with different land types. The prosperity and progress of culture reduces the ESV value of cultivated land to some extent, while the change in land use type leads to the significant loss of ESV in Yinchuan City.

**Keywords:** land use and land cover; ecosystem service value; temporal and spatial patterns; GIS; Yinchuan City

## 1. Introduction

Land use and cover change (LUCC) is a fundamental component of an environmental shift and a major determinant of sustainable development of productivity and human adaptation to global change [1]. LUCC is of great significance for maintaining good ecosystem structure and sustainable development of productivity [2]. LUCCs may also accelerate soil erosion, or destroy agricultural landscapes or natural habitats, which alter biodiversity and makes the ecological environment more vulnerable [3,4]. Population growth and economic expansion are widely recognized as the main drive factors of LUCC [5]. In recent years, interdisciplinary assessment of LUCC status and change has become an increasingly important topic in environmental change studies [6]. With the change in land use, the structure and function of the ecosystem are further affected [7]. Ecosystem is a unified whole that human beings rely on for survival. We live in an artificial ecosystem dominated by cities and farmland [8]. The long-term survival and development of human beings depend on life-sustaining products and services, which are all provided by the structure, process and function of ecosystem services [9]. Therefore, an assessment of ecosystem service values (ESV) helps to identify problems early and to take measures in time so that health and ecological security is promoted [10].

ESV refers to the natural environmental conditions and functions maintained by the ecosystem structure and ecological processes that provide direct or indirect life and guarantee human survival [11]. Ecosystem structure and function are affected and restricted

by land use type, structure and distribution patterns [12]. ESV assessment can quantify and analyze the magnitude of ecosystem service function, which is an important basis for decision-making for regional ecological construction, ecological compensation and policy making. The evaluation model of ESV proposed by Costanza et al. (1997) [13] clarifies the methods and theories of ESV research. With the help of Costanza's theory, the academic community conducted relevant studies in different regions and scales [14]. Xie et al. (2003) [15] determined the equivalent of ESV in China, which is based on development in the country, and laid a foundation for the evaluation of ESV in China. However, in recent years, with the frequent occurrence of extreme weather and the drastic change in land use, the content and mode of ecosystem services have undergone spatiotemporal changes [16], especially in arid areas. It is necessary to study those long-term spatiotemporal changes in arid regions and understand the driving forces behind them and of their effect on ecosystem services.

However, since construction began on the modern city, the change in land use has affected the ecosystem service value [17]. Researching the effects of time and space in Yinchuan City has been rare and has not been part of any of the pilot city designs. As a matter of fact, the Yinchuan example of this lack of urban ecological theory is typical in the west of China, which makes having a systematic in-depth study that much more important [12].

In view of this, the objectives of this paper are to (1) calculate ESV based on LUCC data for Yinchuan City from 1980–2018, (2) to analyze the spatial and temporal patterns of LUCC and ESV, and (3) to discuss the response of ESV to LUCC. This study will help to understand the rational planning of resources in Yinchuan City, promote the improvement and optimal allocation of ecological resources, help realize regional sustainable development and provide a theoretical reference for the construction of other arid cities in Northwest China.

## 2. Materials and Methods

### 2.1. Study Area

Located in the northwest of China and the middle of the Ningxia plain, Yinchuan City lies on the western edge of Ordos, with the Helan mountains to the west. The Yellow River runs through the city, making Yinchuan an important trading town on the ancient Silk Road. As the provincial capital, Yinchuan City has become the economic, political, cultural, scientific research, transportation, military and financial center of Ningxia. It is also the hub city of the China-Mongolia-Russia region, the New Eurasian Land Bridge Economic Corridor and the western window of China. It possesses a very distinct climate, since the Yellow River provides very good surface irrigation conditions and there are strong artificial intervention factors in the ecosystem [18]. Temperatures in the city average 8.6 °C, with average annual sunshine duration at more than 3000 h and annual precipitation at 200~220 mm, and a frost-free period of about 190 days. The territory is rich in water resources, with numerous lakes and wetlands, a diversity of soil types suitable for crop growth. As of 2018, the total area of the city measured 8874.61 km$^2$, and total population was $2.25 \times 10^6$, with a year-to-year GDP growth of 7.2%. Its jurisdiction consists of three districts (Xingqing, Jinfeng and Xixia), two counties (Helan and Yongning) and one county-level city (Lingwu City) (Figure 1). The region possesses a complete ecological system that makes a study in the heterogeneity of a regional natural environment precious, not to mention seeing all this in an economic development context. Due to the poor ecological conditions in the periphery of Yinchuan City, the natural supply and adjustment abilities of the ecological services are weak. As one of 20 designed pilot cities, the landscape pattern and ecological system in the urban planning of Yinchuan would be part of a fundamental research question [19].

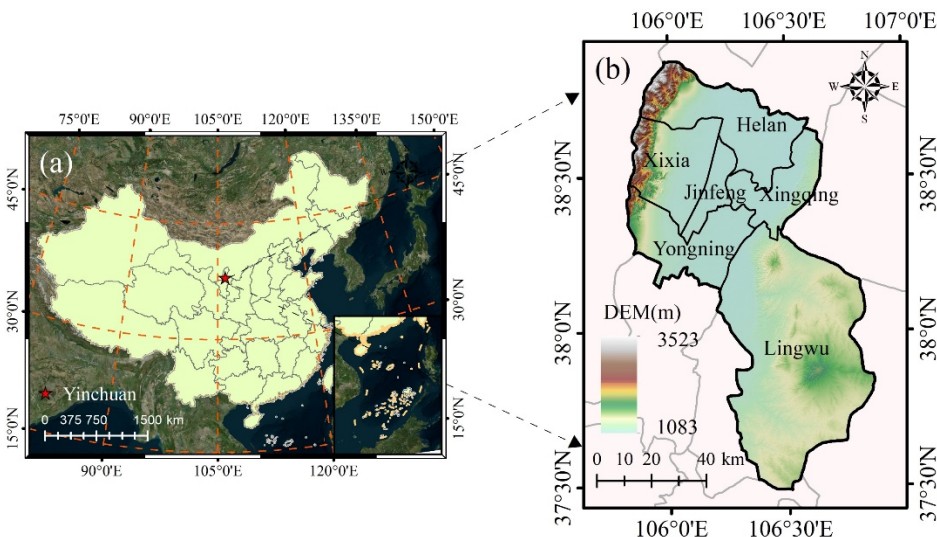

**Figure 1.** (**a**) The geographical location of Yinchuan city in China; (**b**) DEM of Yinchuan city; Location of the study area.

### 2.2. Data Sources

This article uses the land use data products issued by Tsinghua university, the spatial resolution is 30 m (http://data.ess.tsinghua.edu.cn/) (accessed on 9 June 2021) in 1980, 1990, 1995, 2000, 2005, 2010, 2015 and 2018, and the product data reveals a high precision. Eight raster data (1980, 1990, 1995, 2000, 2005, 2010, 2015 and 2018) were selected. According to the characteristics of the study area, the land use types were reclassified into six first-level classifications: cultivated land, forest, grassland, water body, building land and bare land. The digital elevation model (DEM) data were from NASA's SRTM data with a spatial resolution of 30 m in 2002 (http://srtm.csi.cgiar.org/) (accessed on 9 June 2021).

### 2.3. Land Use Change Index

In order to study the change rate and status of a single land use type [20], three indices of land use dynamic attitude (RS), land use spatial dynamic attitude (RSS) and land use change state (PS) were selected.

RS represents the speed and amplitude of land use change [21]. RSS represents the degree of spatial change for a certain land use type. PS represents the change trend and state of a certain land use type [18,22]. Its formula is as follows:

$$RS = \frac{Ub - Ua}{Ua} \times \frac{1}{T} \times 100\% \tag{1}$$

$$RSS = \frac{Uin + Uout}{Ua} \times \frac{1}{T} \times 100\% \tag{2}$$

$$PS = \frac{Uin - Uout}{Uin + Uout} \tag{3}$$

where, *Ua* is the area of a certain ground object type at the beginning of the study, *Ub* is the area of a surface feature type at the end of the study. *T* is the research period; *Uin* is the sum of areas converted from other feature types to this feature type in the research period *T*; *Uout* is the sum of the areas transformed from a certain feature type to other feature types in the research period *T*.

### 2.4. Landscape Metrics

The landscape metrics can reflect landscape heterogeneity and allow for analysis of various ecological processes at different scales, making this index an important piece of the research puzzle in ecological landscape science [23]. Based on the study area and scale

size, patch density (PD), the Shannon diversity index (SHDI), Shannon evenness index (SHEI), aggregation index (AI), landscape shape index (LSI), contagion index (CONTAG) and European nearest neighborhood index (EMN_MN) were selected [24]. PD is a basic index for landscape pattern analysis. It expresses the number of patches per unit area and facilitates comparisons between landscapes of different sizes. SHDI is widely used in community ecology to measure diversity, reflecting landscape heterogeneity and is particularly sensitive to the uneven distribution of patch types in the landscape. The richer the land use and the higher the fragmentation in a landscape system, the higher the SHDI value. SHEI is equal to the SHDI divided by the maximum possible diversity for a given landscape abundance (equally distributed across tessellation types). SHEI = 0 means that the landscape is composed of one type of patchwork and has no diversity: SHEI = 1 means that the patchwork types are evenly distributed and have maximum diversity. When the SHEI value is close to 1, there is no obvious dominant type in the landscape and the patchwork types are evenly distributed in the landscape. AI is calculated based on the length of the common boundary between pixels of the same type. When there is no common boundary between all the pixels of a type, the type has the lowest degree of aggregation; and when the common boundary between all the pixels of a type reaches a maximum, it has the largest aggregation index. CONTAG indicates that the dominant patch types in the landscape are well connected, conversely, the landscape has a dispersed pattern of multiple elements and a high degree of fragmentation. LSI is a measure of shape complexity by calculating the extent to which the shape of a patch in an area deviates from a circle or square of the same size; a larger LSI indicates a more fragmented landscape. EMN_MN is the most common distance metric and measures the absolute distance between points in space. The closer the distances the more similar they are and the more likely they are to interfere with each other [25]. Fragstats version 4.2 software was used to calculate each landscape metrics and, for the specific ecological significance and calculation formula, we referred to the literature [26].

*2.5. Ecosystem Service Value*

Based on eight raster maps for land use data at 30-m resolution, the Xie et al. [15] correction on the Chinese land ecosystem service value was adopted, allowing the study to observe the actual situation of Yinchuan and obtain the value coefficients of five land use types in Yinchuan City. In this paper, the ESV of each land use unit area is calculated based on the research methods of Costanza et al. [13] and Xie et al. [15], using the following equation:

$$\begin{aligned} \text{ESV} &= \Sigma(A_k \times \text{VC}_k) \\ \text{ESV}_f &= \Sigma\left(A_k \times \text{VC}_{fk}\right) \end{aligned} \qquad (4)$$

where, ESV represents the total ecosystem service value (yuan) of Yinchuan City, $A_k$ represents the area (km$^2$) of the land use type K, and VC$_k$ represents the ESV coefficient (Yuan·km$^{-2}$), corresponding to the land use type K. ESV$_f$ represents the total value of individual service of the ecosystem (yuan), f refers to the 11 ecological service types, such as Food production, Raw material production and Water supply in Table 1, while VC$_{fk}$ represents the value coefficient of individual service function (yuan·km$^{-2}$) [27].

**Table 1.** The equivalent coefficient of value per unit area of various land use types in Yinchuan city (yuan·km$^{-2}$) [28].

| Primary Types | ESV Calculation Index | Cultivated Land | Forest | Grassland | Water Body | Bare Land |
|---|---|---|---|---|---|---|
| Provisioning services | Food production | 42.85 | 6.47 | 5.45 | 22.31 | 0.00 |
| | Raw material | 5.18 | 14.65 | 8.01 | 12.43 | 0.00 |
| | Water supply | −71.54 | 7.49 | 4.43 | 185.31 | 0.00 |
| Regulating services | Gas regulation | 34.81 | 48.03 | 28.10 | 45.48 | 0.68 |
| | Climate regulation | 17.99 | 144.10 | 74.26 | 100.32 | 0.00 |
| | Purify environment | 5.31 | 43.60 | 24.53 | 155.85 | 3.41 |
| | Water regulation | 75.97 | 114.12 | 54.33 | 2154.10 | 1.02 |
| Supporting services | Soil conservation | 7.29 | 58.59 | 34.24 | 55.19 | 0.68 |
| | Nutrient cycling | 5.99 | 4.43 | 2.73 | 4.26 | 0.00 |
| | Biodiversity | 6.61 | 53.48 | 31.17 | 177.48 | 0.68 |
| Cultural services | Aesthetic landscape | 2.86 | 23.51 | 13.80 | 112.76 | 0.34 |
| | Total | 133.33 | 518.47 | 281.04 | 3025.48 | 6.81 |

The above calculation's comprehensiveness, and the accuracy of the results, will positively affect methods [13]. On the basis of previous studies, this paper adjusted the equivalent coefficient of value per unit area: the equivalent coefficient of value per unit area of the cultivated land was calculated by weighting the ratio of paddy fields to dry land in the statistics of land use area in Yinchuan City for the span of years from 1980 to 2018; Since most of the forest species in Yinchuan City are shrubs, the equivalent coefficient of value per unit area of shrubs was adopted as forest. The mean value of grassland and meadow was used as the equivalent coefficient of value per unit area of grassland in this region. The average of the equivalent coefficient of value per unit area of the water system and wetland was adopted as water body. The equivalent coefficient of value per unit area of bare land was adopted. Since the equivalent coefficient of value per unit area of building land is 0, the ecological value of this type is not considered in this paper. The equivalent coefficient of value per unit area of each land use type in Yinchuan City are shown in Table 1. The value quantity of standard equivalent factor ESV of the improved ecosystem service valuing method is $3.41 \times 10^5$ yuan·km$^{-2}$ [13,15]. In recent years, the change from cultivated land to building land in Yinchuan has triggered a decline in the value of the water supply, resulting in a negative ESV of cultivated land in the water supply. According to the selection indexes in this paper, 11 second-level ecological service types are selected under the four first-level service categories (provisioning, regulating, supporting and cultural services) of ecological service types [28]. These include food production, raw material and water supply, and constitute provisioning services. Gas, climate, water and purified environment are all part of the regulating services. Soil conservation, nutrient cycling and biodiversity comprise supporting services, while aesthetic landscapes fall under cultural services.

*2.6. Calculation of ESV Change Rate*

Change rate formula of ecological service types:

$$C = \frac{A_j - A_i}{A_i} \tag{5}$$

where $C$ is the change rate of the types of ecological services, $i$ is the starting year and $j$ is the ending year. $A_i$ and $A_j$ are the value quantities of an ecological service type in Yinchuan City in two periods ($10^4$ yuan·km$^{-2}$).

Dynamic attitude calculation formula of comprehensive ecological service types:

$$\text{LC} = \left[\frac{\sum_{i=1}^{n} \Delta \text{LU}_{i-j}}{2\sum_{i=1}^{n} \text{LU}_i}\right] \times \frac{1}{T} \times 100\% \tag{6}$$

where LC is the annual change rate of ecological services, $\text{LU}i$ is the value quantity of category $i$ ecological service types at the starting time of monitoring; $\Delta \text{LU}_{(i-j)}$ refers to the absolute value of the transformation of category $i$ ecological service types to non-category $i$ ecological service types in the monitoring period; $T$ is the length of the monitoring period; When the time period of T is set as year, the value of LC is the annual change rate of ecological services in the study area [29].

Furthermore, this study used the Pearson correlation coefficient method to analyze the correlation between the total value of the four primary ecological service types and the ESV of different land types [30].

## 3. Results

### 3.1. Spatial and Temporal Characteristics of LUCC

Figure 2 shows the spatiotemporal change characteristics of LUCC in Yinchuan City from 1980 to 2018. The area of land use types in Yinchuan City has changed significantly from 1980 to 2018, with the dominant landscape being grassland and cultivated land. The area of grassland decreased by 14.81%, from 2965.52 km$^2$ in 1980 to 2526.45 km$^2$ in 2018. The area of cultivated land increased by 14.69%, from 1892.71 km$^2$ in 1980 to 2218.59 km$^2$ in 2018. In contrast, the area of forest and water body changed little, the former decreased from 527.28 km$^2$ in 1980 to 507.93 km$^2$ in 2018, and the latter increased from 265.38 km$^2$ in 1980 to 271.16 km$^2$ in 2018. The area of building land changed the most, from 212.81 km$^2$ in 1980 to 746.38 km$^2$ in 2018, an increase of 533.57 km$^2$ or 250.73%. It was also found that the area of bare land decreased gradually during the study period. The above analysis shows that in recent years, the degree of land use and development in Yinchuan City has been greatly enhanced and urbanization has expanded. In addition, the change in land use types for different periods varied. The area of cultivated land increased significantly from 1980 to 2000, and then remained stable. The area of building land began to increase significantly in 2010, especially during the period from 2015 to 2018, the fastest growth rate. In conclusion, the land use cover of Yinchuan City changed significantly from 1980 to 2018, the building land saw a large increase, while the cultivated land increased overall, with the grassland and bare land decreasing, and the forest and water body remaining stable.

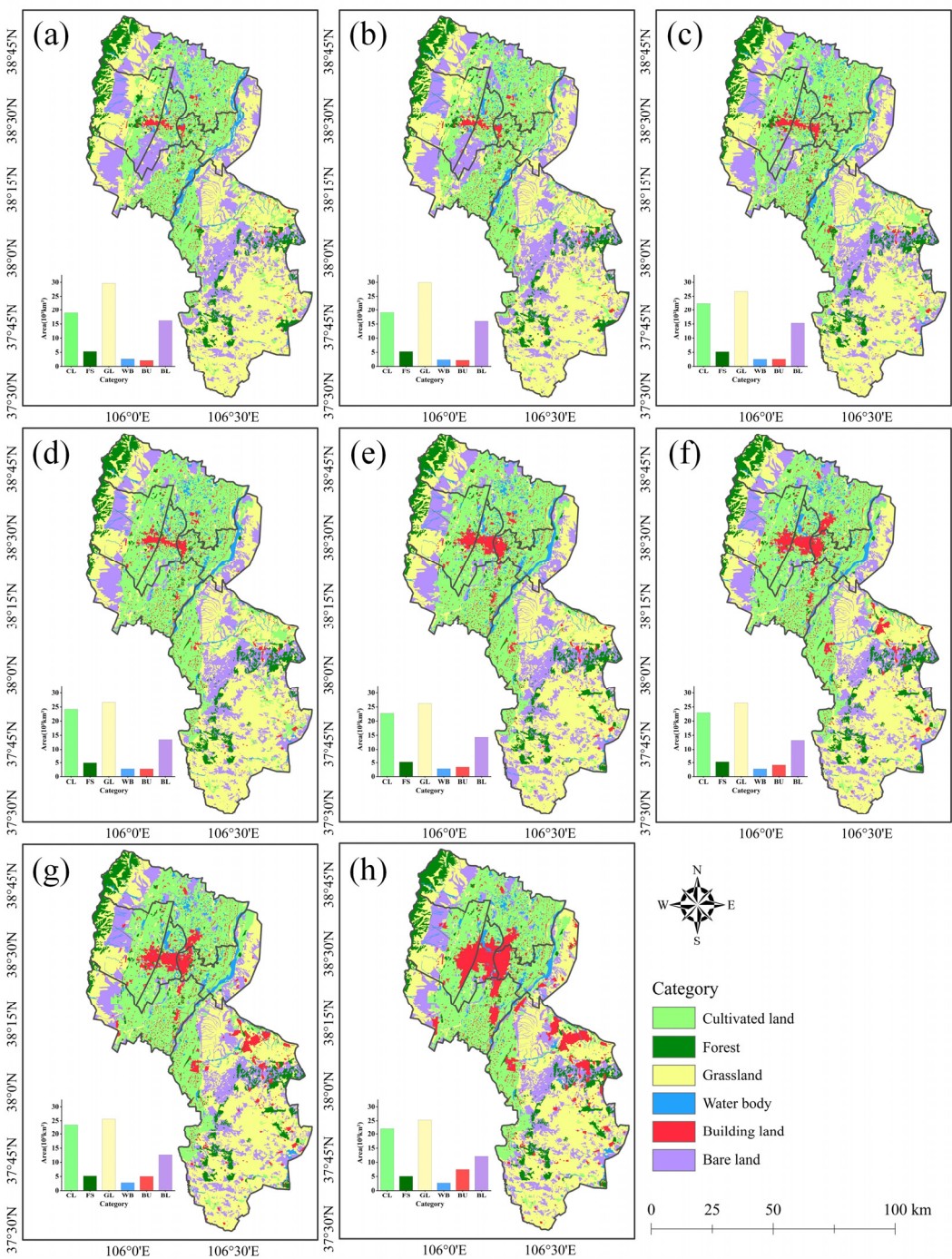

**Figure 2.** The types of LUCC in Yinchuan City from 1980 to 2018. (**a**) 1980, (**b**) 1990, (**c**) 1995, (**d**) 2000, (**e**) 2005, (**f**) 2010, (**g**) 2015, (**h**) 2018.

### 3.2. Spatial Change Characteristics of LUCC

According to Figure 3, there were relatively frequent spatial changes for various land use types in Yinchuan City during different periods from 1980 to 2018. As shown in Figure 3a, most of the cultivated land remained unchanged and was distributed in the north of Yinchuan City. From 1980 to 2000, new cultivated land increased near the farming areas, and from 2000 to 2018, cultivated land expanded outside this area. As shown in Figure 3b, the forest was mainly concentrated in the northwest of Helan county and Xixia district and in the middle of Lingwu City, which did not change significantly during the research period. As shown in Figure 3c, grassland is concentrated in Lingwu City, in the

eastern area of Xingqing district, in the northwest area of Helan county and in the western area of Xixia district. The main source of grassland transfer is cultivated land, and the source of grassland transfer is from the building land in the northeast of Lingwu City and part of the bare land in the middle of Lingwu City. As shown in Figure 3d, the main change in the water area is a small part of the main river that was transformed into forest, while the rest of the area sits relatively unchanged. As shown in Figure 3e, the spatial change in building land is apparent. Only a few areas of building land remain unchanged, while the changed areas are mainly concentrated in the central part of Jinfeng district, the western part of the Xingqing district, the southeastern part of the Xixia district and some areas of Yongning County and Lingwu City. Bare land is the main transfer source of building land. As shown in Figure 3f, bare land has also undergone noticeable spatial changes, with frequent spatial conversion between most areas of Lingwu City and other land use types. In general, land use types in Yinchuan City changed significantly during the 1980–2018 period. This was manifested mainly by grassland and bare land first becoming cultivated land, and then part of that cultivated land becoming building land.

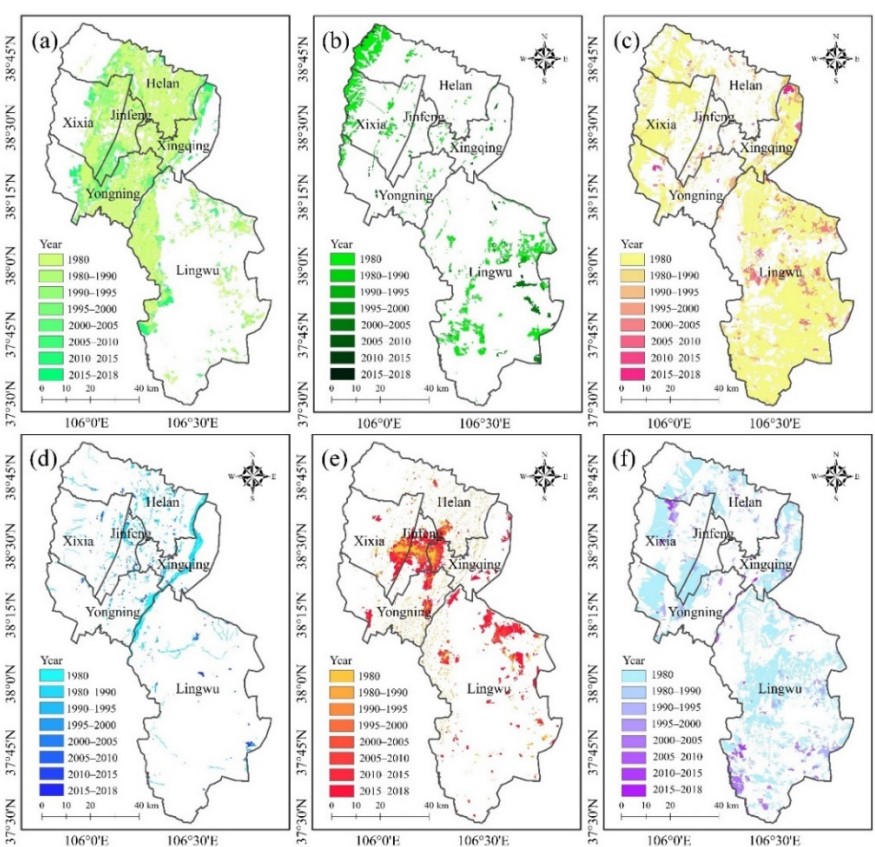

**Figure 3.** Spatial changes of LUCC in Yinchuan city in different time period. (**a**) Cultivated land, (**b**) forest, (**c**) grassland, (**d**) water body, (**e**) building land, (**f**) bare land.

### 3.3. Dynamic Change Characteristics of LUCC

Figures 4–6 show the annual land transfer matrix in the form of RS, RSS and PS of Yinchuan City from 1980 to 2018. In the study period (Figure 4), the absolute value of RS of grassland, cultivated land, building land and bare land was larger, which indicated that the four land use types changed greatly over the past 39 years, while the change in other land use types, such as forest and water body, remained relatively stable. Grassland was basically in a state of reduction at each stage and reached a negative maximum of 2.14% in the 1990–1995 period, with the rate of reduction slowing down significantly in subsequent periods. Bare land, on the whole, showed a downward trend, indicating that human beings began to reclaim large areas of bare land and the development of urbanization was

accelerated. The building land showed a state of growth in all periods. The value of the building land from 2015 to 2018 grew significantly, reaching 16.94%, indicating that the change range of the building land in this period increased, the growth rate had accelerated and the area presented a continuous upward trend. Bare land and water body have been continuously reduced and converted to building land. In this period, the cultivated land was greatly reduced, resulting in the increase of building land. In terms of the use of space, combined with the analysis of RSS (Figure 5), RSS of the use of building land and water body in this region was relatively large from 1980 to 2018, indicating that the spatial transfer area of building land and water body was relatively large, reaching 6.72% and 2.34%, respectively, with the spatial transfer being relatively frequent. The RSS of grassland, cultivated land, forest and bare land was small, which indicates that the spatial transformation frequency of these four types of land features was small. From 1990 to 1995, the RSS of these six types of ground objects changed significantly compared with those of other time periods. RSS indicated the movement in and out of a land use type over a period of time, reflecting whether this land use type switched frequently with other land use types. A small value indicated that there are few transitions between this land use type and other land use types, while a large value indicated that there were frequent transitions between this land use type and other land use types. From the point of view of PS (Figure 6), the PS value of grassland and bare land decreased from 1980 to 2018, while the PS value of cultivated land and building land increased, with the PS value of forest land and water body remaining relatively unchanged. This indicates that the artificial and conscious change of land use type is expanding, due to grassland and bare land being used for the construction of cultivated land and building land. The PS for building land was increasing across periods, indicating that building land consistently increased more than it decreased and that building land has increased significantly. In contrast, the PS of other land types oscillates between periods, which meant that other land types transform each other between periods, but most of the areas were transferred to building land.

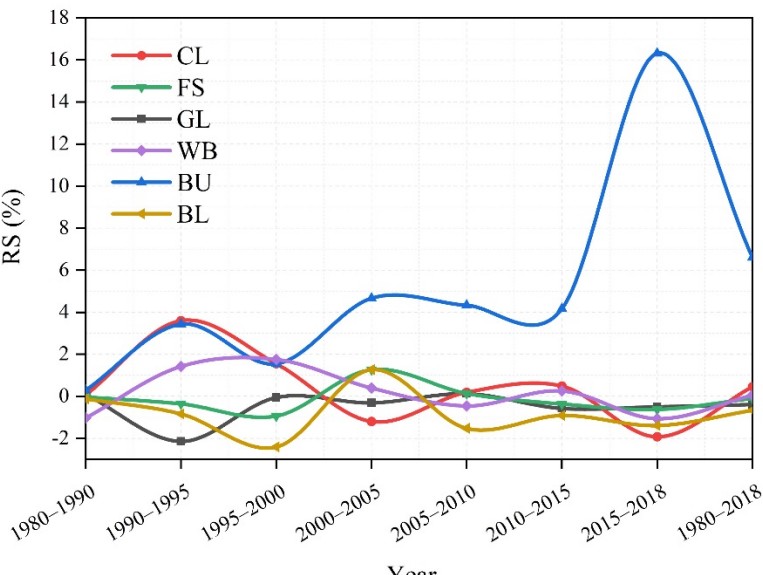

**Figure 4.** The RS of Yinchuan City from 1980 to 2018 (%). cultivated land–CL, forest–FS, grassland–GL, water body–WB, building land–BU, bare land–BL.

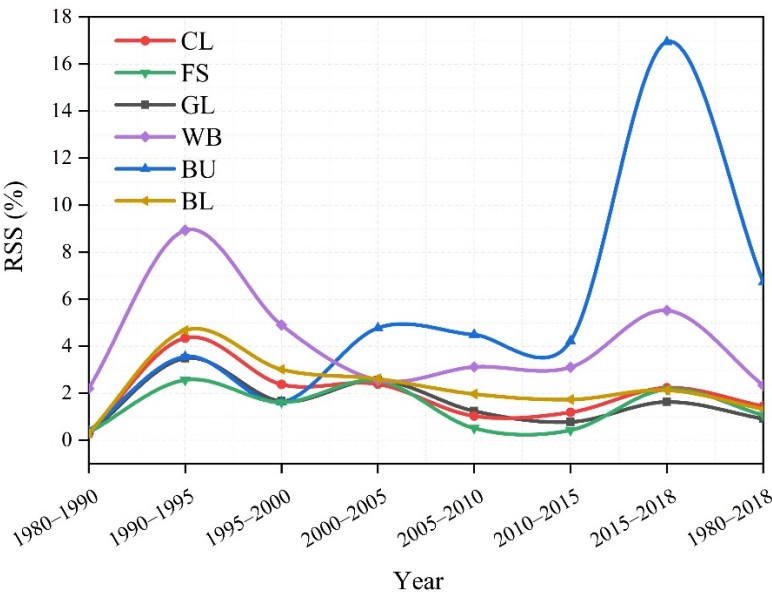

**Figure 5.** The RSS of Yinchuan City from 1980 to 2018 (%). cultivated land–CL, forest–FS, grassland–GL, water body–WB, building land–BU, bare land–BL.

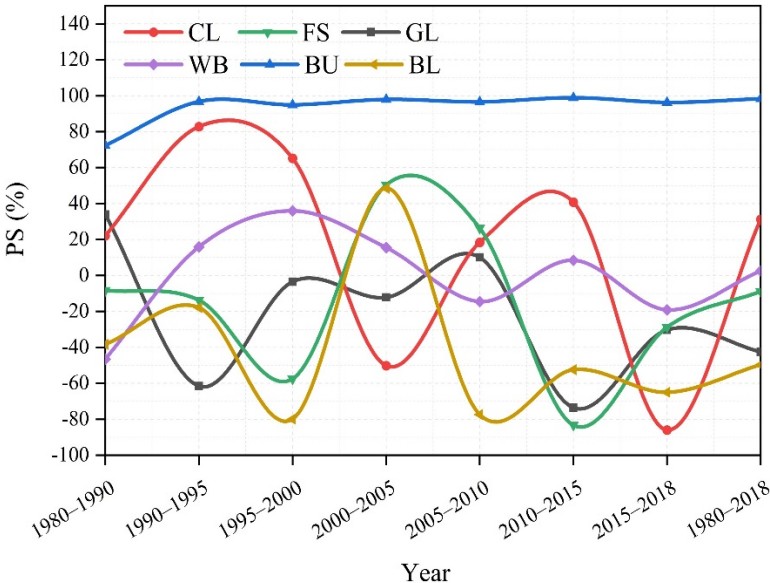

**Figure 6.** The PS of Yinchuan City from 1980 to 2018 (%). cultivated land–CL, forest–FS, grassland–GL, water body–WB, building land–BU, bare land–BL.

### 3.4. Change Characteristics of Landscape Metrics

In combination with Figure 7, PD showed a continuous decreasing trend from 1980 to 2018, indicating that both the number of patches per unit area and the degree of plaque fragmentation decreased. SHDI and SHEI showed the same trend, with an upward direction in the overall trend, which indicates that the land use types in the landscape region were becoming increasingly rich and the landscape types increasingly complex. AI showed an upward trend during the study period, indicating that the degree of non-randomness or aggregation of different plaque types increased and the degree of fragmentation decreased. LSI showed a downward trend, indicating that the landscape patches in the study area became more unidimensional over the past 38 years, showing a large area of patchy distribution. CONTAG showed a downward trend, indicating that the connectivity of different patch types within the landscape decreased during the study period, while the dominant

landscape remained separate from other landscapes. EMN-MN showed an upward trend from 1980 to 2015, indicating that the nearest neighbor distance of different patch types in the landscape increased while the agglomeration degree decreased. The period of 2015 to 2018 saw a significant decline, mainly due to the agglomeration, distribution and area increase of building land, indicating that human activity has been affecting the urban landscape. In general, the SHEI, SHDI, AI and EMN-MN indices of Yinchuan increased from 1980 to 2018, while the PD, LSI and CONTAG indices decreased. The landscape heterogeneity, diversity and evenness increased and the landscape types became more and more complex.

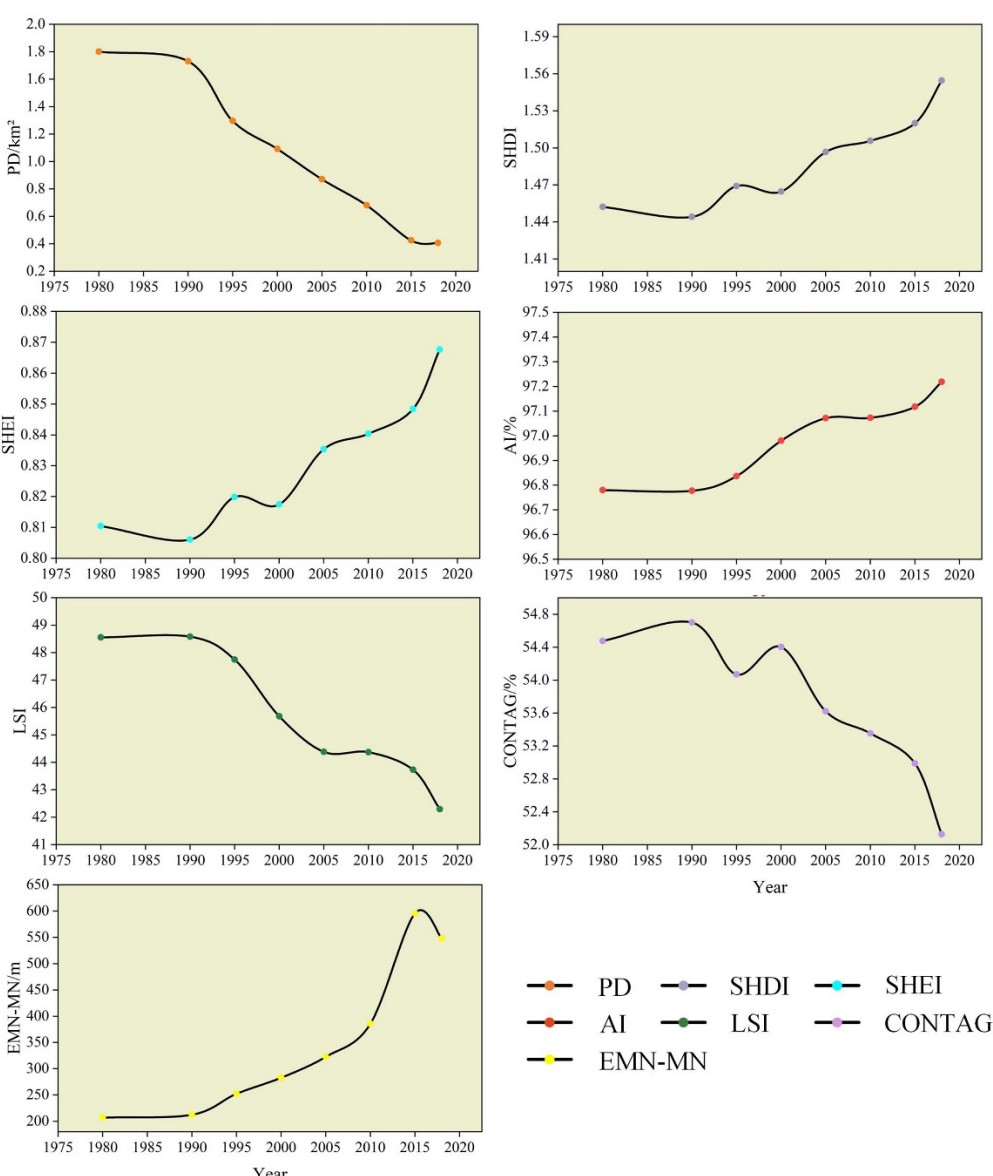

**Figure 7.** The change of landscape indices in Yinchuan City from 1980 to 2018.

### 3.5. Spatial and Temporal Characteristics of ESV in Different Land Use Types

Based on the equivalent coefficient of value per unit area, the ESV of different land types and the ESV of 11 second-level ecological service projects can be obtained according to the area of different land types, as shown in Figures 8 and 9. Figure 8 shows the ESV changed trends for each land type. Among them, the ESV trends of water body and cultivated land are consistent, and the average ESV of water body scores highest among the five types. This is mainly due to the abundance of water resources in the territory, there are

numerous lakes and wetlands with a high value of the equivalent coefficient, causing high ESV in the water body. In addition, there was only a small interannual variation of ESV between forest and bare land, especially for bare land, where the slope of the corresponding line segment was close to 0. However, the ESV of forest showed a state of decrease at first and then an increase around the year 2000. During the study period, the ESV of each land type changed significantly from 1980 to 1995, during which the ESV of grassland, water body and cultivated land changed significantly, with the increases and decreases gradually levelling off. As for the total amount of ESV, it decreased by $0.75 \times 10^9$ yuan from $21.73 \times 10^9$ yuan in 1980 to $20.98 \times 10^9$ yuan in 2018, indicating that the total amount of ecological value provided by various land types in Yinchuan decreased during this period. For a specific land type, forest and bare land remained almost unchanged, while water body and cultivated land showed an upward trend. Meanwhile, grassland showed a downward trend, with the decreasing magnitude larger. In general, the total amount of ecological value of each land type is decreasing, and the ESV of grassland, water body and cultivated land varies greatly, while the ESV of forest and bare land varies slightly. This may be related to Yinchuan's economic development strategy as well as local human activities and natural factors. In addition to using the ESV of each land type for quantitative analysis, the ESV values of corresponding ecological service types in different years can also be obtained, as shown in Figure 9.

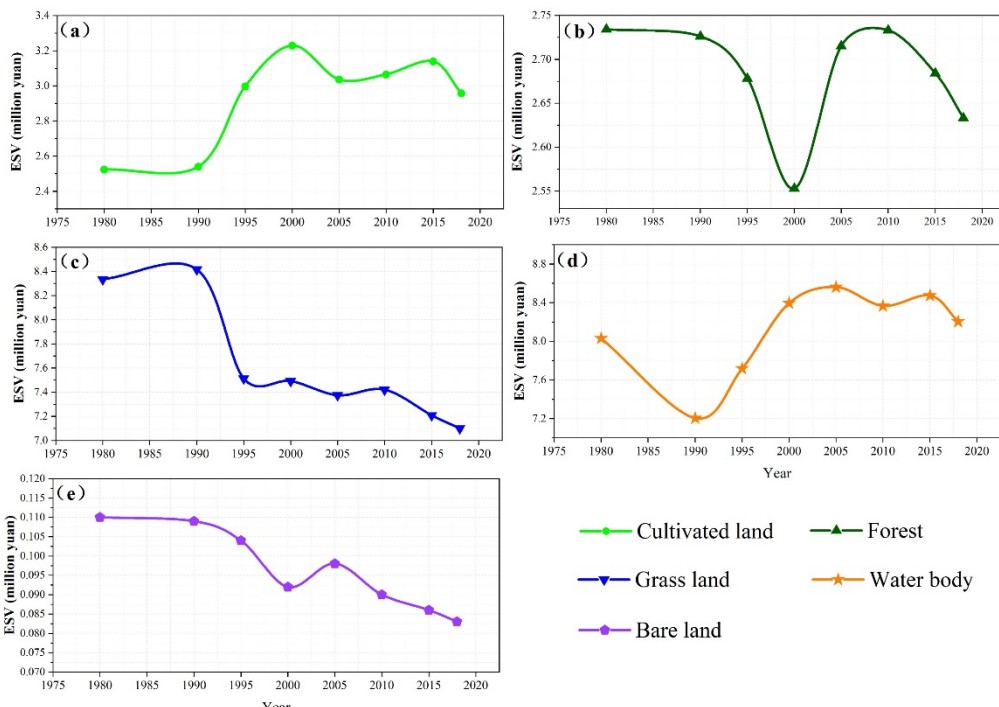

**Figure 8.** ESV values of different land types in Yinchuan City (million yuan). (**a**) Cultivated land, (**b**) forest, (**c**) grassland, (**d**) water body, (**e**) bare land.

Figure 9 shows that ESV values of the corresponding ecological service types varied in different years from 1980 to 2018, among which the ESV values of water supply showed the most significant change, displaying a state of full negative value and reaching its lowest point in 2000. Water regulation accounts for the largest proportion of ESV among all service types, indicating the importance of hydrology to an ecosystem [31]. This was followed by climate regulation, gas regulation, biodiversity and soil conservation. Ecosystem is a complex entity composed of many elements. In order to make its development dynamic and balanced, it is necessary to coordinate and balance all components within it. However, during the study period, ESV values for most of the service types showed a downward trend, indicating that the value brought by the ecosystem to human beings was decreasing

and the degree of human damage to the ecological landscape was increasing. There are also categories where the ESVs, such as food production and nutrient cycling, increase. However, the increase is small and the accompanying growth showing fluctuation, indicating that social and economic development, and human activities, have a significant impact on the secondary ecological service types in Yinchuan [32].

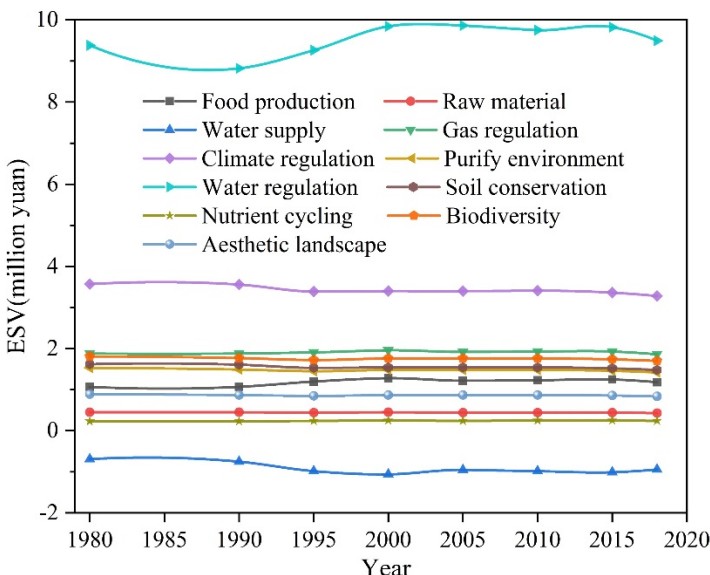

**Figure 9.** ESV values of different ecological service projects in Yinchuan from 1980 to 2018.

In order to more clearly display the change rates in ESV values for various land use types and ecological service types during the 1980–2018 period, the annual rate of change of ESV values for various land use types in Yinchuan City during that period were successively calculated by using the Equation (6), as shown in Figure 10 below.

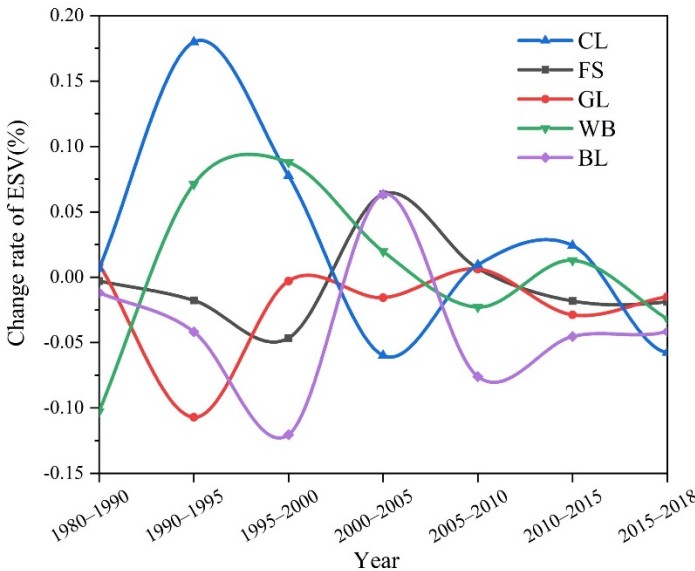

**Figure 10.** Change rate of ESV of different land types in Yinchuan city from 1980 to 2018. cultivated land–CL, forest–FS, grassland–GL, water body–WB, building land–BU, bare land–BL.

Figure 10 shows that the change rates of ESV values of land types varied significantly through the various years. The average annual rate of change from 1980 to 2018 is in the last column of the table. In the study period, cultivated land and water body showed a

downward trend, and the growth rate of cultivated land was much higher than that of water body. Forest, grassland and bare land showed a downward trend, and bare land showed the largest annual reduction rate, followed by grassland, with forest showing the least. During the study period, each land type appeared to buck the trend. In general, bare land and forest showed a downward trend, while cultivated land and water body showed a continuous upward trend. The change rate in grassland fluctuated significantly, and the maximum value reached 10.72. In addition, with the rapid economic development of Yinchuan City, social progress may lead to adverse changes in the ecological environment [33]. The change rates for all five land types showing negative numbers from 2015 to 2018 indicates a downward trend in ESV values for the corresponding land types. Therefore, it's necessary to consider the issue of sustainable development and creating more harmony between social production and ecology, when it comes to Yinchuan City development.

Ecosystem service function (ESV$_f$) values change at different rates in different regions (Figure 11). It's worth noting that the ESV of food production, gas regulation, climate regulation and nutrient cycling decreased significantly. However, the ESV$_f$ of Yinchuan's aesthetic landscape increased significantly. The ESV$_f$ in gas regulation, purify environment, soil conservation, nutrient cycling, biodiversity and other aspects decreased significantly. In general, the service value of multiple service ecosystems showed a downward trend from 1980 to 2018.

Many scholars have proposed different calculation methods for evaluating ESV, but the calculation method proposed by Xie et al. [15], based on the research results of Costanza, has been widely accepted and recognized. Based on the table "Equivalence of Ecosystem Service Value per Unit Area in China", this study calculated the ESV of Yinchuan City for the period from 1980 to 2018, using a 30-m resolution land use type as the regional revision coefficient. For this paper, researcher Yang (2015) [19] compared the calculation results of ESV in Yinchuan City from 2005 to 2010 and found that the change in trend and magnitude were basically the same, indicating that the overall settlement results were reasonable. It shows that the ESV method of using high-resolution land use data revision in this paper is feasible, and the accuracy of the revised ESV data in Yinchuan City has improved accordingly.

In this paper, the ESV correlation calculation method was used to obtain the correlation coefficient indexes between the two variables of five land types and four first-level ecological service types, as shown in Figure 12.

Figure 12 shows a correlation analysis of the four first-level types of ecological services and different land types with their total value calculated. It can be seen that supply, regulation and support service types have a high correlation with different land types, with a high negative correlation between regulation service and different land types. There was also a negative correlation between cultural services and cultivated land, which indicated that, to a certain extent, the prosperity and progress of culture reduced the ESV value of cultivated land. In addition, one sees that there are more negative correlations among the coefficients between cultivated land and different ecological service types. The smallest correlation coefficient is 0.01, indicating that the total value of ecological services has a low correlation with the forest, and that forest does not contribute significantly to the overall ecological services.

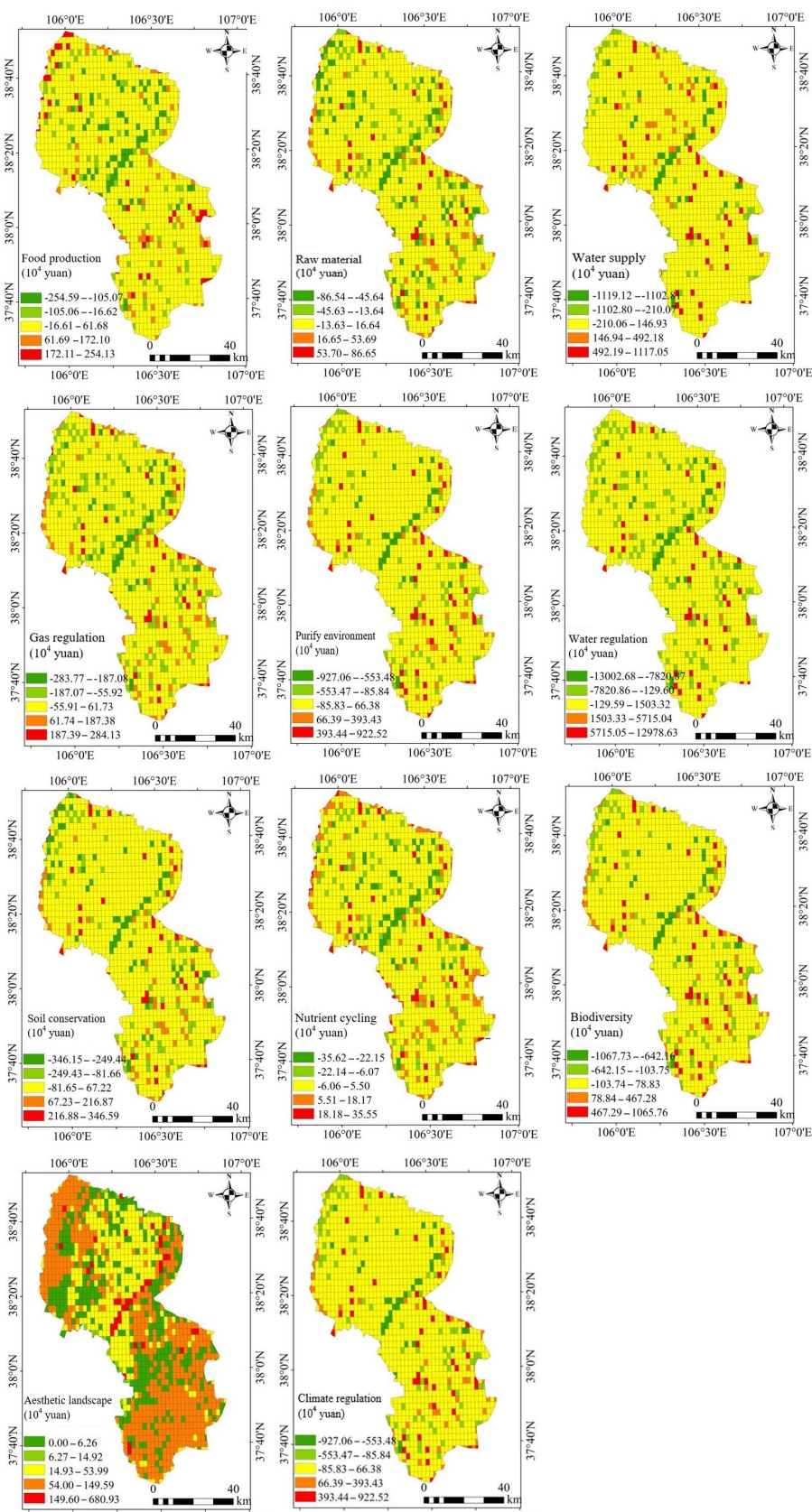

**Figure 11.** Spatial variation of ESV for various service types from 1980 to 2018.

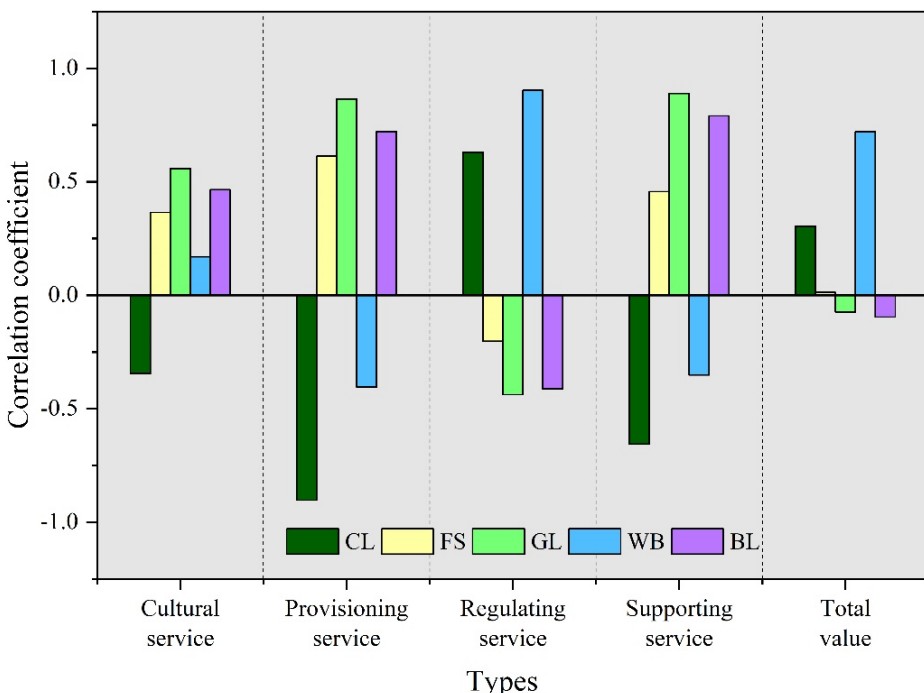

**Figure 12.** The impact of LUCC on ESV from 1980 to 2018. cultivated land–CL, forest–FS, grassland–GL, water body–WB, bare land–BL.

## 4. Discussion

### 4.1. Dynamic Change Characteristics of LUCC

LUCC and Landscape Pattern Change in Yinchuan City changed significantly during the period 1980–2018, with a significant increase in building land, an overall rise in cultivated land, a declining trend in grassland and bare land, and stable forest and water body. The area of building land increased by 538.908 $km^2$, accounting for 7.2% of the total area of Yinchuan (Figure 2). The conversion of cultivated land to building land accelerated after 2015 and the rate of urbanization intensified, mainly due to the corresponding increase in demand for building land as the city developed and its economy and population grew, and the impact of human activities on the development of the urban landscape intensified. The spatial pattern of the landscape in Yinchuan over the past 40 years has been characterized by the transformation of grassland and bare land into cultivated land first, and then part of the cultivated land into building land (Figure 3), probably due to the intensification of human disturbance activities in recent years, which has led to frequent exchanges between cultivated land, grassland, bare land and building land, resulting in significant changes in the spatial pattern of the urban landscape. The seven landscape metrics (PD, SHDI, SHEI, AI, LSI, CONTAG and EMN-MN) can reflect well the spatial variability of the changing landscape pattern in Yinchuan, with the urban landscape developing in a regular and balanced direction (Figure 7).

Studies have shown that LUCC and landscape pattern change affect major ecological processes in an ecosystem, such as energy exchange, water cycles, soil erosion and biogeochemical cycles, which all affect ecosystem services [16,34]. Based on this study, it can be seen that the reduction of grassland reduction and the expansion of building land are the main characteristics of LUCC changes in Yinchuan.

### 4.2. Impact of LUCC and Landscape Pattern Change on the ESV

Based on the value equivalent factor per unit area proposed by Xie et al. (2015) [35], the ESV of various land covers from 1980 to 2018 was calculated, and we found that total ESV decreased by $0.75 \times 10^9$ yuan (Figure 8). These changes are mainly due to the decrease in ecosystem services and the decrease in grassland and building land (Figure 2). Even

though the water body is small, it contributes significantly to ecosystem services and is a key factor in improving regional ecological and ecosystem services [14,16]. Overall, the LUCC type change resulted in a significant loss of ESVs in Yinchuan City. The shift from cultivated land to building land has largely resulted in reduced food production and raw material extraction [36], as well as reduced soil formation, biological control and water supply services. Of the 11 specific ecosystem services studied, 90% declined, with diminishing returns, and most of the services were provided by natural ecosystems (Figure 11).

In Figure 12, the maximum correlation coefficient is 0.90, indicating a high degree of correlation between regulating services and water areas, with water supply playing an essential role in regulating the other services of the ecosystem [37]. At the same time, the relationship between ecosystem value and water area is the highest, followed by cultivated land. It shows that water resources play an important role in population drinking and ecosystem services in Yinchuan City [38]. A rational use of water resources should additionally be promoted, with lower consumption and wasting of water. In addition, cultivated land is the basic guarantee of food security. In addition to its basic function, it also plays an important ecological function. While making full use of cultivated land, it is necessary to maximize the protection of cultivated land to prevent its salinization or turning into unusable land [18]. To have different land types release the maximum benefit of ESV value, there needs to be land and ecological coordination, balanced development, all in the name of building a green Yinchuan foundation.

This is consistent with the results of many studies around the world that show that urban sprawl has a negative impact on the provision of other key ecosystem services [16], such as nutrient cycling, climate and water regulation [39], food production and biological control [40], erosion control and water treatment [41]. In addition, urban expansion also leads to environmental hazards [42]. The government should strictly control the expansion of construction land and pay attention to the development potential of bare land, while effectively protecting the amount of cultivated land. In addition, greater efforts should be made to protect ecological land in grassland, which has a high ecological value coefficient and a greater impact on the value of ecosystem services. A reasonable proportion of the area of this type of land in the study area will play an important role in improving the quality of the ecological environment in the study area.

*4.3. Future Perspectives*

In this paper, we use $30 \times 30$ m land use data, the accuracy of which is subject to further confirmation. Field surveys can improve the accuracy of LUCC. In the future, we can use remote sensing images in combination with field surveys to improve the reliability of studying ESV. In addition, the decline in ESV in Yinchuan is mainly caused by rapid changes in LUCC, which limits the ability of ecosystems to provide sustainable ecosystem services and may lead to long-term degradation of environmental quality. Considering the fragile natural environment of Yinchuan, human activities will have irreversible impacts on ecosystem services in Yinchuan, and we recommend that human activities should be conducted with caution in Yinchuan.

**5. Conclusions**

The lack of a long-term analysis of the system of LUCC and ESV of Yinchuan was analyzed. This was based on the existing land use types and ecosystem services accounting data, combined with change rate, dynamic and comprehensive correlation analysis of Yinchuan during 1980–2018, space-time change rule of LUCC and ESV. The impact of LUCC on ESV was discussed, and the following conclusions were drawn:

(1) The land use and landscape pattern of Yinchuan City changed significantly from 1980 to 2018. The building land increased significantly, the cultivated land increased overall, the grassland and bare land decreased, while the forest and water body remained stable.

(2)     The dominant landscape of Yinchuan is grassland and cultivated land, and the spatial connectivity of building land is continuously improving. SHDI, LSI and CONTAG can well reveal the spatial change characteristics of land use landscape in Yinchuan City, with Yinchuan's landscape developing in a regular and balanced direction.

(3)     From 1980 to 2018, the total ESV in Yinchuan decreased by $0.75 \times 10^9$ yuan; the distribution of ESV was uneven, and the high value area was distributed in the southeast, with Lingwu City as the representative. The central ESV was generally low, represented by Jinfeng district. There were significant differences in ESV in northwest China, with the Xixia region as the representative. The ESV of bare land and forest showed a continuous downward trend, while the ESV of cultivated land and water body showed a continuous upward trend. The change rate for grassland fluctuated significantly, with the maximum value reaching 10.72% (1990–1995).

(4)     The provisioning, regulating and supporting service types have a high correlation with different land types, and the negative correlation between regulating and different land types is the largest. Cultural services were negatively correlated with cultivated land, and the prosperity and progress of culture reduced the ESV value of cultivated land to some extent. In addition, reduced grassland area and increased building area lead to reduced ecosystem services.

**Author Contributions:** Conceptualization, B.W. and T.Y.; data curation, B.W. and T.Y.; formal analysis, B.W.; investigation, B.W.; methodology, B.W.; software, B.W.; supervision, B.W. and T.Y.; writing—original draft, B.W. and T.Y.; writing—review and editing, B.W. All authors have read and agreed to the published version of the manuscript.

**Funding:** This work is funded by the National Scientific Foundation of China (Grants Nos. 41271024) and Ningxia Philosophy and Social Science Program (19NXYCAF17).

**Institutional Review Board Statement:** Not applicable.

**Informed Consent Statement:** Not applicable.

**Data Availability Statement:** No new data were created or analyzed in this study. Data sharing is not applicable to this article.

**Acknowledgments:** The authors would like to thank the United States Geological Survey (USGS) and the Geospatial Data Cloud for providing access to the orthorectified Landsat imagery and ASTER GDEM.

**Conflicts of Interest:** The authors declare no conflict of interest.

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
