# Peer review of "Assessing Impact of Land Use Change on the Ecosystem Service Value in Yinchuan City from 1980 to 2018"

_sustainability, doi:10.3390/su13158311_

Round 1

Reviewer 1 Report

A check of terminology is needed in order to avoid undetermined  or ambiguity in concepts (i.d. structure; sustainable development of productivity at line 30).

Some repetitions should be avoided (i.e. structures is mentioned three times in lise 47, 48, 49); human survival twice in lines 46, 47)r

Too many figures are included in the paper. Many of them could be moved to an Annex.

Geographical description of Yinchuan should be concentred in ch. 2.1, moving lines 62-69.

Indicators at par. 2.4 should be adequately described.

Specific values used for different ecosystem services  (line 140 and following) should be critically described and commented.

Reviewer 2 Report

The manuscript presents an important simulation of the the LUCC data of Yinchuan City from 1980 to 2018, calculated the ESV, analysed the temporal 13 and spatial patterns of LUCC and ESV, and discussed the response of ESV to LUCC. These types of studies are useful for urban planners and decision makers in metropolises. The work must be published; however, I invite the authors to review the suggestions made.

Lines 35 – 37: This section can also be justified with a more current publication, for example: https://doi.org/10.3390/su13116039

Line 100: Maps must be improved. Figure 1.a. China appears alone, it must represent neighboring countries (all of the same color, unlike China). The oceans surrounding China should appear. 1.b and 1.c figures are similar in extension area, should be only one with all the required information. In these maps it is important to represent part of the characteristics described in between lines 84 - 99.

Lines 108-109: Why did the authors use the DEM from NASA's SRTM data with a spatial resolution of 90 m? DEM with spatial resolution of 30 m (1 arc-second for global coverage) is available. Check: https://www.usgs.gov/centers/eros/science/usgs-eros-archive-digital-elevation-shuttle-radar-topography-mission-srtm-1-arc?qt-science_center_objects=0#qt-science_center_objects

Lines 125-133: The Fragstats program must be cited and referenced. I suggest justifying the selection of the landscape metrics, with other recent LUCC studies in which they have used these metrics, for example:

https://doi.org/10.1016/j.scitotenv.2017.12.143

https://doi.org/10.3390/ijgi9100583

Lines 457-489: In the conclusions, repeat part of the discussion. I suggest synthesizing this section, placing exclusively the conclusion of the work.

Reviewer 3 Report

Dear authors,

This research paper describes the actual topic – Assessing impact of land use change on the ecosystem service value in Yinchuan City from 1980 to 2018. Thus, authors using the LUCC data of Yinchuan City from 1980 to 2018, calculated the ESV, analysed the temporal and spatial patterns of LUCC and ESV, and discussed the response of ESV to LUCC. Authors present the the results showing that, from 1980 to 2018, the building land increased significantly in Yinchuan City, as did the  cultivated land.

And I would like to share with authors some remarks too: it seems important to notice that it would be needed to concentrate on the discussion and conclusions of the study. Thus, when developing these sections it would be needed to include to the debate more newest theoretical implications, thus accessing deeper insights.

Reviewer 4 Report

  • This paper presents a case study of determination and evaluation of land cover and land use change in a Chinese region. The data is valid. However, since the manuscript was not neatly prepared, the values of the paper cannot be evaluated highly. There are several typo errors such as hm2 in line 144 and 146, as well as justification and line spacing inaccuracy such as lines 182-193, etc. The methodology is not well described. The results are not linked to the methodology. There is also inconsistency of terminology use such as the one in table 1. The figures are difficult to read and the order of presentation of the land use category is not consistent throughout. The discussion can be further improved, as presently sounds like a summary. 
  • Section 2.3, more reference to literatures on RS, RSS, and PS will be needed to verify the validity of methodology.
  • Section 2.4, a short description of each indicator is needed. Section 3.4 presented the results only. For example, how the two diversity indices are similar or different in definition and what is the implication of the results.
  • From equation (4), VC is the coefficient for each area type, its unit is yuan/km2.
  • Equation 4, f should be explained by linking to the values in Table 1. 
  • The sentence in line 148 to 152 is not understandable. What is the ESV index? It is not defined. Line 156 said Table 1 presents the ESV index. However, this is not consistent with the caption of Table 1: the ESV calculation indicator. Line 291, this number is referred to as the equivalent coefficient of value per unit area. This strongly needs correction.
  • In fact, Table 1 is entirely from [24]. Representing here may need a more specific explanation, otherwise it may be left for reference. Explanation of some interesting values in Table 1 would be useful such as cultivated land in water supply as a negative value, etc.
  •  Table 1, what is the meaning of total as a row summation of value/km2? This is presently not understandable, so it needs more explanation and strong references. 
  • Line 157, what is “the value quantity of standard equivalent factor ESV”?
  • Equation 5, i and j are not properly defined, line 170, 171.
  • Figure 3a to f, the varying colors show the temporal change. The light color indicates that the change was earlier while the dark color indicates that the change was later or more recent. But this section is titled as spatial. Less is discussed about temporal as presented mainly in the (sub) figures. As the degree of change is different for different types of land. Water land is less pronounced while building is more pronounced.  
  • Figure 3, 4 to 6, and 10, the presenting order of land type in the figures should be consistent throughout the paper, for ease of reading.
  • Figure 4 to 6 and 8, it is difficult to identify the line colors. Adding labels to the graphs will be helpful.
  • Figure 4, building land increased during 2015-2018 while other types of land decreased but to a lesser extent. What type of land had changed into building, although it is described in line 257 somehow.
  • Line 249 to 252, more discussion is needed. As RSS combines in and out changes of each land type. What does this mean in a meaningful way, as shown in the figure. How does a small value compare to a large value?
  • Figure 6 is not well discussed. The building land is apparent. But PS of the other lands are swinging over the years. What does this mean?
  • Figure 7a, PD/hm2?
  • Line 287 and 288, more heterogeneity or diversity is expectable in case of a growing city. What is particularly found in this study? Discussion with respect to regional/urban development policy is among several issues to be addressed. Anything will do in addition to those in Section 4.1. 
  • Line 296, water body is high because the unit value from table 1 is high. So it may need further explanation. 
  • Line 309, 310, again this can be naturally expected. 
  • Basically Figure 8 and table 2 are similar, so they should be presented in a similar way, either table or chart. Probably a chart is preferred to a table for the content presented in Table 2. 
  • Line 338, this should be referred to the formula in section 2.6. 
  • How are the values in Table 3 calculated? based on equation 4 or 5? Are the values the yearly change rate? or absolute during the specific period? This is not clear. 
  • Again, table 3 may be better presented in a graphical way. 
  • Around line 374, the correlation analysis was not explained in the methodology. 
  • Typo in line 419, 0.75*10^9.  

Round 2

Reviewer 1 Report

None. It is publishable.

Reviewer 2 Report

In this new review, the authors present several modifications that improved the manuscript. The paper can be published in the present form

Reviewer 3 Report

Congratulations for your efforts to review the article. It's seems better now.

Reviewer 4 Report

Dear authors,

After the reading of the revised manuscript, I’m satisfied with the corrections, as the authors covered all my points, including new explanation and discussions, as well as clarifying the existing ones. The paper is much more complete.

Thus, I recommend the paper for publication after an English proof by the publisher.

Best regards and good continuation of the work done so far.